# Optical Modelling and Phylogenetic Analysis Provide Clues to the Likely Function of Corneal Nipple Arrays in Butterflies and Moths

**DOI:** 10.3390/insects10090262

**Published:** 2019-08-22

**Authors:** Adrian Spalding, Katie Shanks, Jon Bennie, Ursula Potter, Richard ffrench-Constant

**Affiliations:** 1Centre for Ecology and Conservation, University of Exeter in Cornwall, Penryn Campus, Penryn TR10 9FE, UK; 2Spalding Associates (Environmental) Ltd., 10 Walsingham Place, Truro TR1 2RP, UK; 3Environment and Sustainability Institute, University of Exeter Penryn Campus, Penryn TR10 9FE, UK; 4Centre for Geography and Environmental Science, Peter Lanyon Building, Penryn Campus, Treliever Road, Penryn, Cornwall, PenrynTR10 9FE, UK; 5Microscopy & Analysis Suite, Faculty of Science, University of Bath, Claverton Down, Bath BA2 7AY, UK

**Keywords:** compound eyes, Lepidoptera evolution, reflectance, UV light

## Abstract

The lenses in compound eyes of butterflies and moths contain an array of nipple-shaped protuberances, or corneal nipples. Previous work has suggested that these nipples increase light transmittance and reduce the eye glare of moths that are inactive during the day. This work builds on but goes further than earlier analyses suggesting a functional role for these structures including, for the first time, an explanation of why moths are attracted to UV light. Using a phylogenetic approach and 3D optical modelling, we show empirically that these arrays have been independently lost from different groups of moths and butterflies and vary within families. We find differences in the shape of nipples between nocturnal and diurnal species, and that anti-glow reflectance levels are different at different wave-lengths, a result thereby contradicting the currently accepted theory of eye glow for predator avoidance. We find that there is reduced reflectance, and hence greater photon absorption, at UV light, which is probably a reason why moths are attracted to UV. We note that the effective refractive index at the end of the nipples is very close to the refractive index of water, allowing almost all the species with nipples to see without distortion when the eye is partially or completely wet and providing the potential to keep eyes dry. These observations provide a functional explanation for these arrays. Of special interest is the finding that their repeated and independent loss across lepidopteran phylogeny is inconsistent with the explanation that they are being lost in the ‘higher’, more active butterflies.

## 1. Introduction

Recent advances in DNA analysis, computing hardware and phylogenetic software, have facilitated analysis of lepidopteran phylogenies, especially butterflies [1]. Even so, higher-level relationships within the Lepidoptera, and particularly within the species-rich subclade Ditrysia, are still generally far from understood [2]. There is, however, general consensus that the Monotrysia are the most ancient group of Lepidoptera, that in the Dytrisia, the Noctuoidea, Geometroidea and Drepanoidea at least are more advanced than the butterflies, although there is some dispute about the exact order of these families and superfamilies [3,4]. A recent re-classification of UK Lepidoptera [5] places those families relevant to this paper in the following order: Tortricidae, Hesperiidae, Nymphalidae, Lasiocampidae, Saturniidae, Sphingidae, Geometridae, Erebidae, Noctuidae. A robust phylogeny is essential to facilitate the analysis of large-scale environmental and evolutionary processes and patterns exemplified by Lepidoptera.

A large number of the Lepidoptera are diurnal, including many of the most ancestral families and the vast majority of butterflies. Many of the moth families contain both nocturnal and diurnal species (e.g., the Geometridae and Noctuidae), and in some species, the males fly by day, the females by night. Some female moths are wingless, or nearly so, and cannot fly at all. Many nocturnal moths are known for their attraction to light [6]. 

The surface of the lepidopteran eye consists of hexagonal facets, the outer surface of which may be covered with tiny protuberances termed corneal nipples [7,8,9]. These nipples are generally arranged in closely packed lattices where each nipple has six neighbours, but each facet has defective arrays where nipples have either five or seven neighbours known as 5–7 co-ordination defects [10]. These 5–7 co-ordination defects are aligned in rows forming boundaries between domains with perfect structures. These defects could be the result of geometric constraints due to eye curvature (or serve a yet undiscovered purpose in the optical properties of these eyes) and mark boundaries between domain orientations; these defects in the nipple arrays are unlikely to be due to random growth accidents.

Bernhard et al. [11] inspected 361 insect species and found three classes of nipple arrays, defined by nipple height: group 1 had minor protrusions, less than 50 nm high (referred to as protusions rather than nipples); group 2 had low-sized nipples, between 50 and 200 nm; group 3 had larger nipples, about 250 nm. In some species, nipples are absent where the surface consists of numerous irregular pleats [12] or micro ridges [13]. The three groups of nipple were found only among the sister Orders Trichoptera and Lepidoptera [11] (which separated sometime in the Mesozoic [14]), These are just one subset of a rich diversity of insect corneal surfaces found in insects which can be categorized as nipple-like structures, maze-like nano-coatings, dimples and parallel strands; the Lepidoptera have been shown to have a mix of nipples (irregular or highly ordered) and parallel strands [15].

Several theories have been put forward as to the function of these nipples. The most popular is that nipples reduce reflection [9] and, therefore, provide increased light transmittance by potentially increasing visual efficiency through increased photon capture for a given stimulus condition [16]. The anti-reflective structures of moth and butterfly eyes have bio-inspired engineering applications (e.g., in the photovoltaic and optical industries [17]). Another theory [18] suggests that corneal nipples mainly function to reduce the eye glare of moths that are inactive during the day, so to make them less visible for predators. A third theory suggests that nipples reduce adhesion due to the decrease in real contact area between contaminating particles and the eye surface [19]. Lastly, it has been suggested that the reduction or loss of corneal nipples in most butterflies is concomitant with a vanishing trait associated with evolutionary development [1]. More recently, it has been shown that different categories of nanostructures can be seen in various insect groups without any correlation between the predominant type of nanocoating and the evolutionary advance of the group, the categories replicating a diverse set of Turing nanopatterns [15].

In this paper, we provide evidence concerning whether differences in the length and slope of corneal nipples correlate with the evolutionary development of the different lepidopteran families or with other characteristics such as diurnal or nocturnal behavior, sexual dimorphism, anti-glow reflectance, response to light at different spectra and wetting of the eye. 

## 2. Materials and Methods 

### 2.1. Experimental Material

Specimens of Lepidoptera from the UK were captured or found in the personal collection of AS. *Hypolimnas misippus* (L.) (Lepidoptera: Nymphalidae) and *Antheraea jana* Stoll (Lepidoptera: Saturniidae) are from Asia, *Alcides zodiaca* Butler (Uraniidae) from Australia. Eyes of old specimens were examined under the microscope to check whether they had deteriorated with age or damaged. Old specimens (caught in 1965) of *Poecilocampa populi* (L.) (Lepidoptera: Lasiocampidae) were at first discarded under the mistaken impression that the eyes had deteriorated with age and replaced with freshly caught specimens, but they were found to be in the same condition. Twenty-two species were obtained from a variety of families representing five diurnal and 17 nocturnal species. The moths *Lasiocampa quercus* (L.) (Lepidoptera: Lasiocampidae) and *Saturnia pavonia* (L.) (Lepidoptera: Saturniidae) have diurnal males and nocturnal females. 

### 2.2. Electron Microscopy

The corneal nipple arrays were studied using standard electron microscopy using eyes severed from heads of dead specimens. Eyes were dissected from insect heads and scales surrounding them were removed. Eyes were fractured in half with one half used for imaging the surface of the eye, the other for imaging a cross-section through a lens. Samples were mounted on SEM stubs using carbon tape and silver paint. The samples were sputter-coated with gold for surface imaging in a JEOL JSM6480LV SEM, or sputter coated with chromium for imaging in a JEOL JSM6310F FESEM.

### 2.3. Optical Modelling 

The simulations were carried out using a multilayer model simulating the graded refractive index layers going from the top of the nipples to their base (Figure 1). The effective refractive index was calculated using effective medium theory and the calculated volume fraction [17,18]. Effective medium theory (EMT) is a method commonly used to represent a repeating structure with sub-wavelength dimensions as a homogeneous medium [17] (Figure 1c–f). Finite-Difference Time-Domain (FDTD) is a more robust method of analyzing many properties of nanostructures but requires far more computational power and time. EMT, although not effective for many direct measurements, is accurate for calculating the effective refractive index. This method allows monte-carlo-based optical modelling (Breaults ASAP in this study) to be used, which drastically reduces computation time for complex models and allows large areas of nanopatterns to be analyzed. This style of modelling has been used elsewhere for predictions of effective refractive index and reflectance from nanostructures [17,18]. 

For the volume fraction calculations, the refractive index of corneal nipple material was taken to be 1.52, following the methods used by Stavenga et al. [18], where the value is taken from interference microscopy results of moth corneal lens samples carried out by Vogt et al. [20]. In theory, if the nipple material is predominantly made of Chitosan, then this follows other direct measurements made in the literature [21]. The other common material found in lepidoptera exoskeletons is Chitin (refractive index of 1.61 [21]), which is physically stronger than Chitosan but not as optically transmissive [22]. Chitosan dissolves in acidic solutions [22] and from the investigations of Kaya et al. [23], the nipple patterns of dragonfly compound eyes are completely dissolved in acidic cleaning processes when isolating the Chitin for measurement [22]. It is difficult to isolate optical properties within nanostructures of natural substances. In fact, this is a clear challenge in measuring directly the refractive index and reflectance of moth eyes whilst isolating parameters (e.g., incidence angle, nipple spacing, nipple width and height) to see their effects and possible evolutionary drives. This is one reason for adopting optical modelling, to allow nipple structure parameters to be investigated independently and to clearly understand their optical effects. Taking the refractive index of 1.52 for the corneal nipple material [20] also allows results to be directly comparable to previous results such as that carried out by Stavenga et al. [18]. 

It should be noted that any inaccuracy is more likely to come from the averaging and shape representation of the measured nipple patterns rather than the actual modelling methods. The optical theory behind nanopatterns, effective medium theory, reflectance, Fresnel equations, and multilayer refractive indexes are described in detail elsewhere [17] and are in line with standard anti-reflective coating modelling, which commonly utilizes nanopatterns such as those found in moth eyes [17].

The nipple shapes in this study were represented by dome tops merging into straight lines towards the base (Figure 1a) to represent the different nipple height groups; the volume fraction of nipple material to air between one nipple axis to the next nipple axis was then calculated with the ratio of chitosan (*n* = 1.52) and air (*n* = 1). The height and width of the nipples were chosen as representative dimensions within the range of the actual moth eye nipple patterns measured via Scanning Electron Microscope (SEM) images of the moth eyes. For these simulations, which focus on the comparison of the nipple heights, the distance between adjacent nipple axes was taken to be 200 nm (Figure 1c–e) and the maximum width of the nipples was also 200 nm, such that the nipples bases are touching (or almost touching) for group 3 (200 and 300 nm height nipples (Figure 1a,c,d). Nipple density or spacing between nipples was not investigated as nipple distances have been found to vary only slightly in the Lepidoptera [18]. However, for lower height nipples, (group 2 nipples and group 1 ridges), the widths were typically shorter and small gaps more common between nipple bases (Figure 1a,e) and hence, the volume fraction is smaller at the base of the nipple such that the bases do not touch. The dimensions and shape of the nipples were taken from SEM images and the radius (half width) of a nipple taken as the parallel width of the nipple at the junction between the bottom of the dome top and linear descent to the base as shown by the inset image in Figure 1a. Figure 1b shows the effective refractive index calculated using this method for the different nipple heights and accompanying half widths (R) (Figure 1a,b). 

The multilayer model of varying refractive index layers (giving a gradient refractive index as shown in Figure 1c–f) was simulated in Breaults ASAP software [24] to give reflectance measurements (Figure 1f). This simulation assumed normal incident light with a divergence angle of ± 0.27° to match that of the sun, originating from a thermal source similar to the sun with a full range of weighted wavelengths for realistic sunlight properties [25]. All real material has a refractive dispersion relationship, where the refractive index of a material changes depending on the incident wavelength. This was able to be included in the simulation software using the refractive index dispersion of chitosan [21] but extrapolation had to be done for wavelengths out with the available dispersion data (250–1750 nm) [21] which may have introduced a small systematic error in these outer wavelengths. Since any errors such as this are applied to all nipple pattern heights investigated, the comparative conclusions will still hold. 

When dealing with subwavelength geometry such as the nipple arrays, multilayer modelling as outlined is a well-known method [26,27,28,29,30,31]. There is much discussion on the accuracy of these models, depending on the specific structure and composition under analysis [28,29,30,31,32,33,34,35,36,37], but the specific method was chosen as it has previously been used for moth eye nano-nipple array patterns by Stavenga et al. [18] and other investigations considering very low (typically termed anti-reflective) surface structures [27]. Furthermore, when comparing the results to Stavenga’s results [18], our results over the visible range match particularly well to Figure 6c in Stavenga et al. [18] and are similar in form to reports by Han et al. [17].

### 2.4. Statistical Modelling 

Differences between sexes and between diurnal and nocturnal species in nipple heights, nipple top area and the height/area ratio of nipples were tested, controlling for phylogenetic effects at the family level, using phylogenetic generalised least squares (PGLS). A phylogeny was constructed to family level based on Rainford et al. [38], and expected correlation matrices under a Brownian model were calculated using the ape v.5.1 package [39] in R [40]. A mean value of nipple height and nipple top area was calculated for each species/sex combination. To calculate top/height ratio, the mean height and top area measurement for each sex/species combination was calculated, and a single aggregate ratio for each species was derived. Sample size for each analysis differed as some species/sex combinations did not have both area and height measurements. Effects of both diel activity pattern and sex were tested simultaneously as fixed factors.

## 3. Results

### 3.1. Nipple Height Classes 

The dimensions of the nipple arrays vary among the species studied, both in the length of the nipple and the spacing between the nipples. Heights vary between either no height/rudimentary height with ridges or protrusions of less than 10 nm upwards to 350 nm (Figure 2). We have classified them into three height groups—Group 1 < 10 nm, Group 2 10–150 nm, Group 3 > 150 nm. For three species, there were two classes of nipple height: for *Deilephila elpenor* (L.) (Lepidoptera: Sphingidae), 13 of the 33 heights measured for females formed a separate (larger) class ($\bar{x}$ = 224.5 nm) vs. a mean of 111.6 nm for the other class and for males nine of 26 specimens tested possessed a mean of 213.9 nm vs. 88.1 nm; for *Epiphyas postvittana* (Walker) (Lepidoptera: Tortricidae), there was no difference between the males but for females there was a separate smaller class for 14 heights out of 34 heights of 84.4 nm vs. a mean for the other class of 230.3; for *Ectropis crepuscularia* ([D.& S.]) (Lepidoptera: Geometridae) there was no difference between the males but for females, there was a separate larger class for six heights out of 21 ($\bar{x}$ = 243.6 nm). 

The height classes used here are different from those used by Bernhard et al. [11] and subsequently, by Stavenga et al. [18] (0–50 nm; 50–200 nm; > 200 nm), as there appears to be a clear division between group 2 and 3 at 150 nm except for *Lithosia quadra*, where nipple length ranges from 123 to 169 nm. 

For group 1, there were two sub-divisions. The nipples were present but very short in *Angerona prunaria* (L.) (Lepidoptera: Geometridae) but absent in *Lymantria monacha* (L.) (Lepidoptera: Erebidae), *Poecilocampa populi*, *Euthrix potatoria* (L.) (Lepidoptera: Lasiocampidae), *Lasiocampa quercus*, *Saturnia pavonia*, *Coscinia striata* (L.) (Lepidoptera: Erebidae), *Calliteara pudibunda* (L.) (Lepidoptera: Erebidae) and *Euproctis similis* (Fuessly) (Lepidoptera: Erebidae). 

### 3.2. Taxonomic Order 

Figure 3 shows species arranged in taxonomic order; species within families are not arranged in any particular order. All families where more than one species was sampled contain species in more than one nipple group except for the Lasiocampidae and the Lymantriinae (both in group 1). The Nymphalidae contain nipple groups 2 and 3 even though both species are within the Nymphalinae. The Saturnidae contain both groups 1 and 3, even though both species are within the family Saturniinae. The Geometridae contain both groups 1 and 2, even though both species are within the Ennominae. Within the Erebidae, the Arctiinae have a species in all three groups. (Only a single species was sampled from the Tortricidae, Hesperiidae, Sphingidae, Noctuidae and Uraniidae). There is no correlation with the evolutionary development of the different lepidopteran families, and no obvious conformity across the families except possibly for Lasiocampidae and Lymantriinae. 

### 3.3. Diurnal and Nocturnal Species

After controlling for phylogeny, no significant effects of diel activity pattern on nipple height (*n* = 46, 25 species) or height/area ratio (*n* = 13, 8 species) were found (Figure 4). Diel flight activity was found to have a significant effect on top area (*n* = 22, 13 species, *p* = 0.018). Night flying species had larger nipple areas by 7450 nm^2^.

Nipples are absent in the species *Lasiocampa quercus* and *Saturnia pavonia*, where the males are diurnal, the females are nocturnal.

### 3.4. Male–Female Differences

After controlling for phylogeny and nocturnality, no consistent significant effect of sex was found on nipple height (*n* = 46, 25 species), nipple area (*n* = 22, 13 species) or height/area ratio (*n* = 13, eight species). 

### 3.5. Reflectance

Reflectance is reduced proportionately with reduced gap and taller nipples, except in visible light and the moth eye receptor range (~300–750 nm) [41,42], where reflectance is greatest for the longest nipples (Figure 5). The relationship between moth eye reflectance and moth eye absorption or sensitivity also depends on the structure of the eye beneath the nipples and in particular, the refraction into these structures. Reflectance is least at the UV end of the spectrum (< 380 nm as shown in Figure 5) for all classes (100 nm = class 2; 200 nm and 300 nm = class 3) (Figure 5). There are clear differences between the nipple height classes, with reflectance least in class 3 in UV light but least in class 2 in visible light (specifically 250–500 nm). It should be noted that wavelengths < 280 nm are mostly, if not all, absorbed by the earth’s atmosphere but that these UV wavelengths are still present from man-made lighting for which some moths are drawn to, perhaps for the reason they are not adapted to such strong light signals of those UV wavelengths (~300–380 nm).

The reflectance of a completely flat material of refractive index 1.52 was also simulated as a reference along with flat chitosan (Figure 5), which has a refractive index dispersion effect on reflectance as shown in Figure 5. The reflectance from the simulated nipple patterns tends towards these references reflectance at high wavelengths as, with respects to the wavelengths, the nipple patterns tend towards a flat surface (wavelength >> nipple dimensions).

### 3.6. The Refractive Index

The effective refractive index at the end of the dome-shaped tops of the nipples is very close to the refractive index of water (1.33) [20] (Figure 1). 

## 4. Discussion

### 4.1. Is the Loss of Nipple Arrays Associated with Evolutionary Development in the Lepidoptera?

In most modern taxonomic classifications, so-called macro-moth species (e.g., families Lasiocampidae, Saturnidae, Sphingidae, Geometridae, Uraniidae, Erebidae and Noctuidae) are generally considered more advanced than the butterflies [2]. It has been suggested [18], since moths are probably ancestral to the diurnal butterflies, that the reduced size of the nipples of most butterfly species indicates a vanishing trait, especially in the Papilionidae, which have virtually no nipples. The current research in contrast, strongly suggests that this is simply a sampling artefact due to the concentration on butterflies in previous studies, as we here clearly show that nipples are also lost in groups of moths, in some at least of the families Lasiocampidae, Saturnidae, Geometridae and Erebidae, but are present in the Nymphalidae and Hesperiidae. Nipples are also present in the nocturnal Hedylidae [43], which started diversifying from other butterfly families around 29 million years ago, in a secondary reversal from diurnality to nocturnality [1]. Further research will surely reveal whether nipples have been lost in all the Lasiocampidae. It is clear that the most ancestral families such as the Micropterigidae and Nepticulidae [44] have well-developed nipples, as do the Incurvariidae [12], which are an ancestral family with close links to the Trichoptera evidenced by the presence of hairs on the hindwings of the males [45]. The Trichoptera are commonly considered a sister group to the Lepidoptera [14] and include both species with full-size (group 3) nipples [11,13] and species (e.g., the Limnephilidae) with no nipples [15]. 

We know that evolutionary transformations from superposition into apposition optics may have occurred several times independently in various microlepidopteran taxa [46]. We show that corneal nipples have appeared and disappeared several times during lepidopteran phylogenetic development.

The lack of knowledge of lepidopteran evolution over geological time scales due to the relatively sparse and poorly studied fossil record [4,47,48] makes understanding more difficult. However, corneal nipples are assumed to have a function. The question remains why many species of Lepidoptera either do not possess any such corneal surface specializations at all, or instead possess micro-ridges radially diverging from the apex of the corneal dome toward the edge of the corneal lens [13]. The question is not to find evidence which supports a particular theory but to outline alternative explanations and then eliminate some by empirical experiment [49].

### 4.2. The Length of Corneal Nipples Varies within Familes

We show that in many of the families, sampled species are found containing more than one nipple group. It is therefore clear that there are other drivers of nipple development than taxonomic development, and that eye morphology cannot be used in cladistic analysis, thus refuting the hypothesis that nipples are a vanishing trait associated with evolutionary development. 

### 4.3. Corneal Nipples are Shorter, More Rounded and with Larger Top Areas in Nocturnal Species

Night flying species have larger nipple areas; this may provide better light reception between nipples, with less incidental light. The length of the nipples is also important. Their efficiency in increasing the amount of light admitted to the eye is height-dependent [13] and we, therefore, might expect that nocturnal species would have longer nipples than diurnal species due to the need to be active in lower light conditions. Indeed, it is often assumed that the reason why certain butterflies are active during the day is because their nipple structures have gradually lessened or even disappeared [17]. However, after controlling for phylogeny, we found no significant effect on nipple heights, although all of the five diurnal species sampled had long nipples (2 in group 2, 3 in group 3). Hence, the absence of a significant effect may be a product of small sample size (nipples are absent in day-flying moth species such as *Coscinia striata* (L.) (Lepidoptera: Erebidae). The shorter nipples, which give a low height to top area ratio, do not absorb as much wide-angled incident light but may be better for absorbing light in one direction, allowing the moth to focus light coming head on, which would be beneficial in low-light conditions. In contrast, night-flying moths may benefit from shorter, shallower and slightly rounder nipples to see small direct but probably weaker images of light more easily in the dark. The long nipples found in diurnal species subjected to high levels of diffuse radiation during the day absorb light from wider angles, perhaps allowing the individuals to more readily see predators; in particular, the smallest moths such as the Nepticulidae and the Micropterigidae have lower visual acuity and sensitivity as moth eyes decrease with body size and these moths may have shifted to diurnal activity to take advantage of the associated higher illumination levels [46]. There is no obvious correlation between the nipple length of the nocturnal species sampled and their flight to light, a finding that warrants further investigation. The case of *Lasiocampa quercus* and *Satonia pavonia* is interesting. To our knowledge, the complete lack of corneal nipples has hitherto not been linked with the fact that in both species the males are diurnal, females nocturnal. (It is interesting to note that the Papilionidae which have short, virtually non-existent nipples [11,18] are generally only active in bright light conditions). Possibly, in both moth species, the females were originally diurnal but evolved to fly at night, perhaps to avoid predators. In the highly sexually dimorphic moth *Orgyia antiqua* (L.) (Lepidoptera: Erebidae), the sedentary, wingless female possesses a much smaller and less well-organized compound eye than the winged males (which are active by day) [50]; both sexes have corneal nipples and any differences in these would be worthy of investigation.

Divergence of males and females in their behavior and habitat preferences, leading to differences in eye form and function, may be relatively recent [51].

### 4.4. There is Less Reflectance and More Light Absorption Where Nipples are Longer

Reflectance is generally reduced proportionately (for UV and IR light) with reduced gap and taller nipples (Figure 5), arguing that there is an optimum curvature and configuration with gap to which the moth eyes conform (reflectance is likely to be highest for Class I nipples at the 0–10 nm range). Despite the fact that many authors suggest that reduction in reflectance is a major factor for Lepidoptera, especially in nocturnal moths [18], we posit that this is an incidental by-product of increased nipple height. For example, cryptic Lasiocampidae species such as *Euthrix potatoria*, which is well-camouflaged, lack nipples despite hiding from predators during the day and it has been found [11] that most of the aposematic species in the family Amatidae (which do not need to hide during the day) were found to possess full-sized nipples. On the other hand, some cryptic nocturnal species such as *Luperina nickerlii* (Freyer) (Lepidoptera: Noctuidae) have long nipples and hide in the open during the day, but can be found at night by eye-glow reflected from observer torches [52]. The use of industrial designs derived from nipple-like structures to reduce reflection [53,54,55] and the antireflective nanoprotuberance present on the wings of Ithomiine butterflies [27] and Sphingidae [56] do not prove that reflection reduction is the cause of increased nipple height. Indeed, eyeshine is visible in the moth *Phalaenoides tristifica* Hübner (Lepidoptera: Noctuidae), despite it being both diurnal (when reflectance would not be problem) and having 250 nm-high nipples, characteristic of nocturnal moths [57]. In addition, reduced light reflectance is not restricted to eyes with regular nipple structures but is also observed in irregular arrangements [10].

### 4.5. Light Absortion is Greatest at the UV Range of the Spectrum (< 380 nm)

The attraction of moths to light is well known, especially light with high ultraviolet (UV) emission [52,58,59,60] and it has been shown that different families are attracted differentially to different wavelengths [7]. Even diurnal insects may be attracted to UV light [61]. Despite this, the attraction to light has never been properly explained, with various theories expounded based on external factors such as moths using light sources for navigation [62], becoming dazzled [63] or the so-called Mach’s band effect where moths see a dark stripe around a bright light [64]. Here, we suggest that the structure of the eye and in particular, the corneal nipple arrays facilitate the attraction to UV light. Within the moth eye receptor range (300–750 nm [F]), reflectance is greatest in visible light and lowest at the UV end of the spectrum (< 380 nm), especially for moths with nipple lengths of 100 nm (group 1); at this wavelength, proton absorption would be greatest, which may explain why moths are attracted to UV light as they would perceive such wavelengths more intensely than other wavelengths. It, therefore, might be expected that the following species (i.e., with nipples ± 100 nm long) would be especially attracted to light: *Epiphyas postvittana*, *Ochlodes sylvanus* (Esper) (Lepidoptera: Hesperiidae), *Deilephila elpenor*, *Ectropis crepuscularia* ([D. and S.]) (Lepidoptera: Geometridae), *Lycia zonaria* ([D. and S.]) (Lepidoptera: Geometridae) and *Euclidia mi* (Clerck) ([D. and S.]) (Lepidoptera: Erebidae). Of these, *Ochlodes sylvanus* and *Euclidia mi* are diurnal and *Lycia zonaria* has a wingless female, but the other species and male *Lycia zonaria* come very readily to light. Further investigation is clearly necessary. 

### 4.6. Is the Closeness of the Effective Refractive Index at the Nipple Top to the Refractive Index of Water a Chance Feature?

Corneal nipples create an interface with a gradient refractive index between that of air and the facet lens material, because their distance is distinctly smaller than the wavelength of light [18] and they act as an impedance transformer equalizing by gradual transition of the refractive index of air to that of the cornea, increasing the transmittance of visible light [11]. We here show that the effective refractive index at the end of the dome-shaped tops of Class 2 and 3 nipples is very close to the refractive index of water. Our observations suggest that by matching the refractive index of water, the role of nipples is to reduce distortion of images that would be expected when water droplets sit on the nipple surface e.g., when vapor condenses into droplets (fogging) on the eye structures so as to reduce visibility by light scattering and reflection. Class 1 nipples do not have this property and it is likely that water droplets would be spaced across and cover the eye surface. Class 2 and 3 nipples would allow a moth or butterfly to see even if part of the eye was wet, e.g., from rain droplets. The wettability of the eye would allow the water to slide off easily; if there was a thin layer of water left, it would have little effect on sight because it would cover only the top of the nipples due to water tension and the closeness of the nipples (the penetration depth of water has yet to be investigated). It has been previously shown that non-closely packed nipples of the mosquito, *Culex pipiens* L. (Diptera: Culicidae) prevent microscale fog-drops from condensing on the ommatidia surface, whereas closely-packed nipples prevent drops being trapped in the voids between the ommatidia, even when the hairs surrounding the eyes are covered with water droplets [65]. This can be compared to the self-cleaning ability and ultra-low water adhesion properties of leaves of the lotus, *Nelumbo nucifera* Gaertn (Proteales: Nelumbonaceae), which removes dust and particles by the moving of water droplets [66]. The match with the refractive index of water at 1.33 seems too much of a coincidence to be insignificant in such an intricate pattern. The refractive index of water decreases very slightly with temperature increase, so changes in temperature would make very little difference to this function [67]. 

## 5. Conclusions

This study suggests a new and hitherto unsuspected biological role of corneal nipple arrays in the Lepidoptera. We show empirically that corneal nipples have both appeared and disappeared several times during lepidopteran phylogenetic development and vary within families. We observed differences in nipples between nocturnal and diurnal species, and that the effective refractive index at the end of nipples is very close to the refractive index of water, with implications for enhanced visual perception and self-cleansing. Lastly, we reveal the existence of different anti-glow reflectance levels at different wave-lengths with reduced reflectance in the compound eyes of Lepidoptera, and hence, greater photon absorption, at UV wavelengths of light and thereby strongly believe that this has a physiological basis, and indeed is the reason why moths are attracted to UV light. Our observations provide a functional explanation for such arrays. 

## Figures and Tables

**Figure 1 insects-10-00262-f001:**
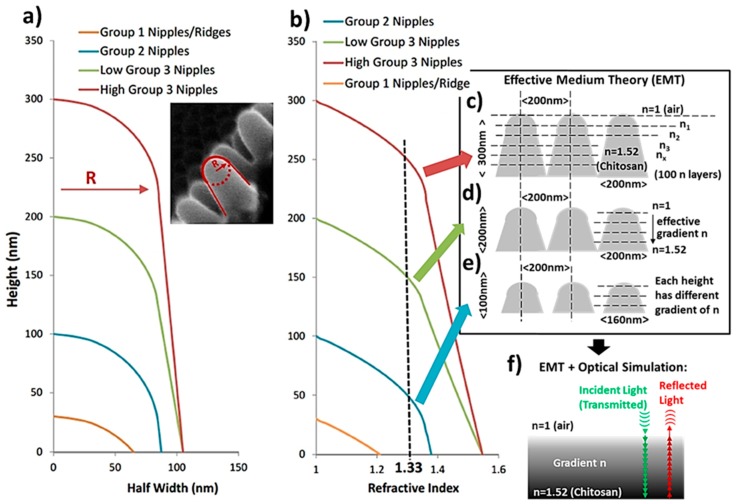
(**a**) Geometric profile representation of the three different groups of nipple height with half widths (Radius, R) and an example SEM inset image showing why a circular dome top going into a straight line is representative. (**b**). Graph of effective refractive index as a function of height (related to width of nipple shape and narrowing spacing between adjacent nipples) showing the refractive index of water. (**c–e**) nipple pattern dimensions and effective refractive index layers which tend towards a unique gradient refractive index for each height. (**f**) Representation of simulation where incident light waves only see the layers of effective refractive index resultant from effective medium theory calculations. Nipple material taken as 1.52 (chitosan), air *n* = 1.

**Figure 2 insects-10-00262-f002:**
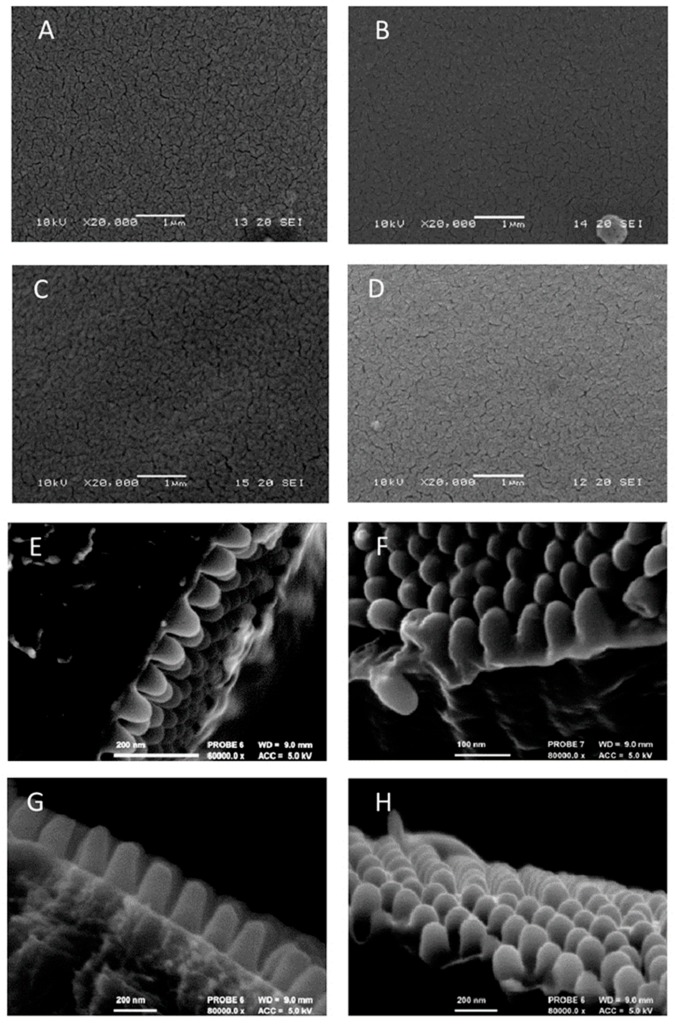
Examples of corneal nipple arrays. (**A–D) no nipples (Group 1)** (**A**,**B**) *Calliteara pudibunda* is a nocturnal species, (**C**,**D**) *Saturnia pavonia* has a nocturnal female (**C**) and a diurnal male (**D**). (**A**) *Calliteara pudibunda* Female ×20,000 (**B**) *Calliteara pudibunda* Male ×20,000 (**C**) *Saturnia pavonia* Female ×20,000 D Male ×20,000. **(E,F) Mid-range nipples (Group 2)** (**E**) *Lycia zonaria* is nocturnal, (**F**) *Euclidia mi* is diurnal. (**E**) *Lycia zonaria* male ×60,000 (**F**) *Euclidia mi* male ×80,000. (**G,H) Long nipples (Group 3)** (**G**) *Spilosoma lubricipeda* is nocturnal (**H**) *Alcides zodiaca* is diurnal. (**G**) *Spilosoma lubricipeda* ×80,000, (**H**) *Alcides zodiaca* ×80,000.

**Figure 3 insects-10-00262-f003:**
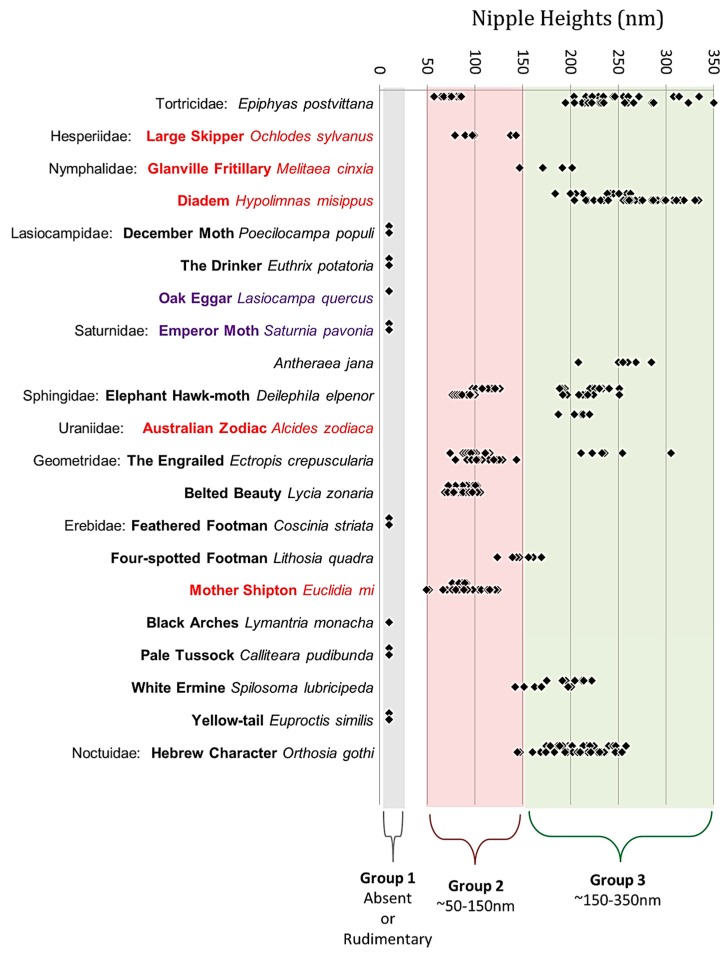
Corneal nipple height data in three groups arranged according to taxonomic order. Nocturnal species are in black, diurnal species in red; species where the males are diurnal and the females nocturnal are in purple. Each point represents a measurement of single nipple height.

**Figure 4 insects-10-00262-f004:**
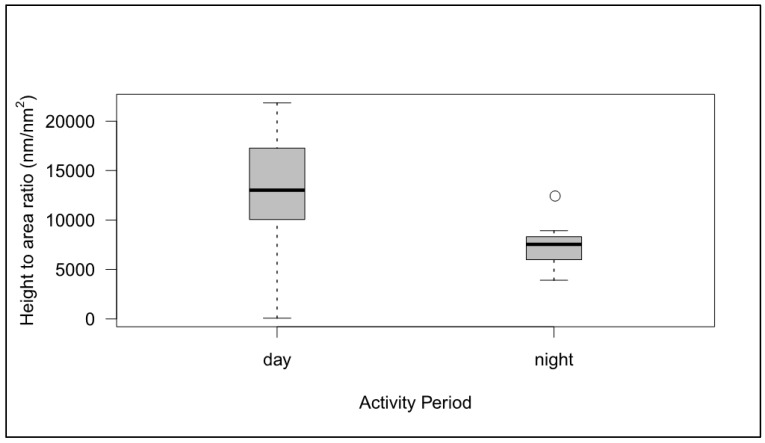
Height/area ratio of species against nocturnal or diurnal activity period for eight species (four nocturnal, four diurnal (*p* = 0.0837) (the nocturnal species are *Deilephila elpenor*, *Ectropis crepuscularia*, *Orthosia gothica*, *Spilosoma lubricipeda;* the diurnal species are *Lycia zonaria*, *Ematurga atomaria*, *Hypolimnas misippus*, *Ochlodes sylvanus*).

**Figure 5 insects-10-00262-f005:**
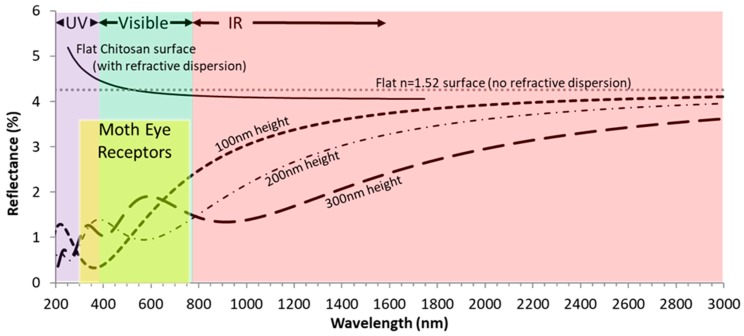
Simulated reflectance of three nipple heights with dimensions and effective refractive index layers as described in Figure 1. As references, the reflectance of a flat surface of refractive index 1.52 is also shown and of flat chitosan following its refractive dispersion relationship. Reflectance is generally reduced proportionately with reduced gap and taller nipples.

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
