# Peer review of "Optical Modelling and Phylogenetic Analysis Provide Clues to the Likely Function of Corneal Nipple Arrays in Butterflies and Moths"

_insects, 2019, doi:10.3390/insects10090262_

Round 1
Reviewer 1 Report
Title: “Optic modelling and phylogenetic analysis….butterflies and moths”
Author: Adrian SPALDING et al.
Novel ides and new hypotheses are infinitely more useful than articles that only confirm what has been known already. It’s in this context that I highly recommend this manuscript by Spalding et al. for publication, irrespective as to whether their hypothesis will ultimately become established as a theorem or not. I do, however, have some comments that I’d like the authors to consider.
I the abstract I’d prefer to see them write “…builds on but goes further than earlier analyses, suggesting a…” Singling out one reference, e,g, Stavenga et al., could suggest bias or favouritism, given that other similar analyses (even cited later in the ms like Dewan et al 2012, for example) exist. Moreover, it is not common practice to include references in an abstract. I wholeheartedly agree with their last sentence of the abstract.
Regarding keywords I’d like to see ‘compound eyes’ included and ‘corneal nuipples’ removed (as the latter are already part of the title).
Lines 46-48: totally correct!
Lines 49-52: some female moths are wingless and don’t fly at all. This could be mentioned here, just like some moths that are parthenogenetic with males being unknown could (cf., Honkanen & Meyer-Rochow 2009).
L62-70: I realize the authors need to provide an introduction to corneal nipples, and mention absence and presence of the latter. Should in that context not also be mentioned that some moths are known to possess microridges or groves rather than nipples on the corneal surfaces and that the heights of these ridges correspond to those of the more abundant nipples?
Having used interference microscopy myself to measure refractive indices of insect eye dioptric structures, how confident are the authors that theoretically derived refractive indices are realistic values?
Results: Microridges are mentioned on page 9, but can anything more be said about their possible evolutionary history in view of the authors’ analyses?
Lines 184-187: what about the tiniest of moths? They possess corneal nipples, but seem mostly diurnal. If indeed corneal nipples facilitates the acceptance of UV and eyes are very small (because of the small size of the insect and its head), then I’d argue small eyes even in diurnal species have to have nipples in order to accept sufficient light to see by.
Discussion, lines 248-275: here a discussion that very small eyes would benefit from nipples could be included, and UV preference ‘pushes’ such moth species to become diurnal as Fischer et al. 2014 conclude. A question not addressed by the authors is the degree to which the curvatures of the eye as a whole and those of individual ommatidia affect the optical performance of the eye (with and without nipples or microridges).
Conclusion: Whether the new ideas stand the test of time will be interesting. At the moment, however, the challenge is there and the new hypothesis provides food for thought and calls for action. What more can you want of a research paper?
References: Why are the titles of some papers written with capital letters (e.g., Horridge et al. 1977; Fischer et al. 2013; Rainford et al. 2014, etc)? In Lau & Meyer-Rochow the species Orgyia antiqua needs to be in italics. The reference (14): Dey 2007 is incomplete and should be dropped as there are many more comprehensive studies than that one by Sudip Dey.
Figures: They are all good and necessary, but in Fig. 2 I should like to see white lines separating the individual micrographs and the size of the figure enlarged to fill the while page.
Reviewer 2 Report
See attached.

Reviewer 3 Report
First, I apologize that my comments may include some irrelevant ones, since I am neither an entomologist nor a (bio-)physician. I hope some will be useful to improve the manuscript. This study examined presence or absence and size ranges of the corneal nipple arrays in various lepidopteran species and estimate the light reflection on the nipple arrays based on the optical modeling of several types of arrays. Combination with phylogenetic analysis showed that the nipples have appeared and disappeared several times in the lepidopteran lineages and refuted the hypothesis that nipples are a vanishing trait associated with evolution. The authors discussed that that reduction in reflectance would be an incidental by-product of increased nipple height. They also suggest that one of the functions of the nipples is the reduction of distortion of images when water droplets sit on the nipple surface as, estimating the refractive index at the tops of the nipples is close to those of water. This article provides some new ideas on the functions of the corneal nipple array, and I recommend the publication after appropriate revisions.
To estimate the gradient of the refractive index on the structure, the refractive index of the material (chitin?) that form nipples should be necessary. Are there any measurements of the material? Please show the value, method, and representative reference.
I could not well understand the parameters in the estimation of the reflectance. Consider to add a schematic drawing of the section of the nipple array used in the simulation for 100, 200, 300 nm in height that correspond to Fig 5. (See Fig. 2c, f in [16] for reference.) The author should explain the incident angle of the light (º), interval of the nipples (nm)and arrangement of the nipples (grid or hexagonal), as well as the height (already shown). These parameters affect the reflectance. Although I do not know the examples in the insect research, but those in marine animals are found in Hirose et al. (J. Mar. Biol. Assoc. UK95:1025–1031, 2015) and Sakai et al. (Zool. Lett. 4:7, 2018).
UV-lights shorter than 280 nm are absorbed by the ozone layer and do not reach the ground, although Figure 5 ranges from 0 to 3000 nm. The author should mention this elsewhere.
Line 146: Are there any reasonable explanations for the grouping based on nipple height, i.e., group 1–3?
Figure 2: Because scale bar is shown in each figure, magnification need not shown in the figure legends. Brightness and contrast should be adjusted for E and G, appropriately.
Figure 3: Consider to add phylogenetic tree based on [31] and order the species according to phylogenetic relatedness. Indicate diel/diurnal/nocternal types.
Line 204: I cannot understand what ‘UV end’ means. Show it as wavelength.
Figure 5. Consider to add the curve of 0 nm height (flat surface) as a control. Clarify this is a real measurement or a simulation based on an optical model.
Line 256–263: Consider to summarize this part in a table.
Line 305–307: Flat surface (0 nm height) should be compared with the 100-nm nipple array in Figure 5.
Minor comments and corrections were directly added in the manuscript pdf.

Round 2
Reviewer 2 Report
The revised version of the paper is much better and publishable in this journal.